# Accurate multi-population imputation of *MICA*, *MICB*, *HLA-E*, *HLA-F* and *HLA-G* alleles from genome SNP data

**Silja Tammi**[1]*, **Satu Koskela**[1,2], **Blood Service Biobank**[2], **Kati Hyvärinen**[1], **Jukka Partanen**[1,2], **Jarmo Ritari**[1]

**1** Finnish Red Cross Blood Service, Research and Development, Helsinki, Finland, **2** Finnish Red Cross Blood Service, Blood Service Biobank, Vantaa, Finland

* silja.tammi@bloodservice.fi

**Data Availability Statement:** 1000 Genomes cohort: Sequence data is publicly available at https://ftp.1000genomes.ebi.ac.uk/vol1/ftp/data_

## Abstract

In addition to the classical HLA genes, the major histocompatibility complex (MHC) harbors a high number of other polymorphic genes with less established roles in disease associations and transplantation matching. To facilitate studies of the non-classical and non-HLA genes in large patient and biobank cohorts, we trained imputation models for *MICA*, *MICB*, *HLA-E*, *HLA-F* and *HLA-G* alleles on genome SNP array data. We show, using both population-specific and multi-population 1000 Genomes references, that the alleles of these genes can be accurately imputed for screening and research purposes. The best imputation model for *MICA*, *MICB*, *HLA-E*, *-F* and *-G* achieved a mean accuracy of 99.3% (min, max: 98.6, 99.9). Furthermore, validation of the 1000 Genomes exome short-read sequencing-based allele calling against a clinical-grade reference data showed an average accuracy of 99.8%, testifying for the quality of the 1000 Genomes data as an imputation reference. We also fitted the models for Infinium Global Screening Array (GSA, Illumina, Inc.) and Axiom Precision Medicine Research Array (PMRA, Thermo Fisher Scientific Inc.) SNP content, with mean accuracies of 99.1% (97.2, 100) and 98.9% (97.4, 100), respectively.

## Author summary

The major histocompatibility complex (MHC) region on chromosome 6 significantly influences disease risk, particularly in autoimmune conditions. To improve fine-mapping of potentially causal genetic variants within this region, we developed accurate imputation methods for inferring functional allelic variation from MHC SNP data. While existing tools primarily focus on classical HLA genes, our study extends to non-classical HLA genes (*HLA-E*, *HLA-F*, and *HLA-G*) and MHC Class I Chain-Related MIC genes (*MICA* and *MICB*) which have specific functions in innate and adaptive immunity. Leveraging population-specific Finnish and multi-population 1000 Genomes references, our imputation models demonstrate high accuracy. Moreover, we tailored models for two widely used genome SNP arrays: the Infinium Global Screening Array (Illumina, Inc.) and the Axiom Precision Medicine Research Array (Thermo Fisher Scientific Inc.). These freely

collections/1000_genomes_project/data/ and SNP genotype data at https://www.cog-genomics.org/plink/2.0/resources#phase3_1kg. Finnish cohort: Genotyping and non-classical HLA/MICAB typing data are stored in the Blood Service Biobank, Vantaa, Finland. Researchers may apply for access to data (https://www.veripalvelu.fi/en/biobank/for-researchers/) The imputation models, trained using the HIBAG algorithm, are available at GitHub (https://github.com/FRCBS/HLA_EFG_MICAB_imputation). R code for training and validation of the imputation models are available in Github (https://github.com/FRCBS/HLA_EFG_MICAB_model_training).

**Funding:** This study was supported by research grants from the Finnish Funding Agency for Technology and Innovation (TEKES, currently Business Finland, https://www.businessfinland.fi/en/for-finnish-customers/home) for the Salwe GID (Personalized Diagnostics and Care) program (ID 3982/31/2013), the Research Council of Finland (for JP, grant 288393, https://www.aka.fi/en/), Cancer Society of Finland (for JP and JR, https://www.cancersociety.fi/), and VTR funding from the Government of Finland. The funders had no role in study design, data collection and analysis, decision to publish, or preparation of the manuscript.

**Competing interests:** The authors have declared that no competing interests exist.

available, multi-population models empower researchers to explore genetic MHC associations in more detail and contribute to our understanding of immune-related disease mechanisms.

## Introduction

The human major histocompatibility complex (MHC) on chromosome 6 is the most gene-dense and polymorphic region of the human genome with many genes related to the immune system. In addition to their role in antigen presentation, the genes in MHC are the most important single factors in transplantation matching and genetic susceptibility to autoimmune and infectious diseases, with more than 400 associations described so far [1,2].

Although the classical HLA genes are considered to explain most of the genetic risk conferred by the MHC, non-HLA variation may also contribute to the risk. The non-classical HLA and MHC Class I Chain-Related (MIC) genes of the MHC class I region function in both the innate and adaptive immunity and have been targets of candidate gene studies showing putative, though conflicting, roles in diseases. Expression levels and polymorphisms in *MICA*, *MICB*, *HLA-E* and *HLA-G* have been associated with autoimmune diseases [3–14], infections [15–23] and susceptibility to cancer [24–33]. There is also growing evidence that *MICA*, *MICB* and *HLA-G* function as transplantation antigens and that their polymorphism and matching may be associated with outcomes of transplantation [34–37]. Moreover, genetic association studies focusing on single nucleotide polymorphisms (SNPs) in the MHC region have identified polymorphisms in the class I region outside of the classical HLA genes that are associated for example with hematopoietic stem cell transplantation (HSCT) outcomes and risk for graft-versus-host disease (GVHD) [38–40], as well as autoimmune diseases [41–43]. As a result of the tight linkage disequilibrium (LD) in the MHC segment, these SNPs may not be the actual causal variants.

Computational methods such as SNP2HLA [44], HLA*IMP [45], HIBAG [46], CookHLA [47] and DEEP*HLA [48] have enabled SNP-based inference of the classical HLA alleles from large genotyped study cohorts. Imputation has proved to be a valuable tool in finding classical HLA disease associations in large biobank-scale cohorts, but it is of note that the accuracy of the tools strongly depends on the algorithm, reference size and population, and SNP density and coverage [49–51]. For example, using population-specific reference data can help increase the accuracy of the imputation of classical HLA alleles [52]. On the other hand, multi-population references support studies of more heterogeneous cohorts [53–55].

Although many imputation tools have been developed for the classical HLA, to our knowledge, only a few studies have included the non-classical HLA and other genes in the MHC [56,57]. As far as we know, these imputation references are not publicly available. Accurate imputation of these genes would facilitate association studies in large population cohorts and enable fine mapping of the MHC associations outside of the classical HLA to better understand the role of MHC variation in disease etiology. In the present study we construct and validate a high-accuracy imputation method for *MICA*, *MICB*, *HLA-E*, *HLA-F* and *HLA-G* using both population-specific and multi-population 1000 Genomes references using the HIBAG ensemble classifier algorithm. The models are available in GitHub (https://github.com/FRCBS/HLA_EFG_MICAB_imputation) and can be readily applied for allele imputations in local genotype data collections.

## Results

### Allele calling

Sequencing data for the Finnish reference was obtained through targeted PacBio long-read sequencing (FIN I) or full MHC genome sequencing (FIN II). *MICA*, *MICB*, *HLA-E*, and *HLA-F* alleles were assigned at two-field resolution (*i.e.*, defining protein sequence-level variation) and *HLA-G* alleles at four-field resolution (*i.e.*, defining nucleotide sequence-level variation), since the majority of genetic variation impacting *HLA-G* expression and regulation resides in the untranslated regions of the gene [58]. The final reference dataset comprised 761 samples for *MICA* and *MICB*, 441 for *HLA-E*, 211 for *HLA-F*, and 435 for *HLA-G*.

To obtain *MICA*, *MICB*, *HLA-E* and *HLA-F* allele typings for the multi-population reference, the 1000 Genomes phase 3 short-read whole exome sequencing (WES) data from 3906 samples were analyzed in two-field resolution. However, the high-resolution *HLA-G* was not feasible to type with exome reads. Many of the samples had no sequencing reads in *MICA*, *MICB*, *HLA-E* or *-F* gene area or had too low read depth for the analysis and thus had to be excluded. Additionally, samples with inconclusive typing results, ambiguous typing results due to phasing ambiguities, or possible novel alleles were excluded to ensure sufficient quality of the reference. Since the read depth of the data was low (average median read depth 69x and average lowest read depth 13x), read depth threshold 10x was used in the analysis. Of the analyzed samples, 1,555 *MICA*, 1,606 *MICB*, 2,037 *HLA-E*, and 1,293 *HLA-F* typing results were included in the final reference.

### Imputation model development

We trained and cross-validated imputation models for *MICA*, *MICB*, *HLA-E* and *HLA-F* in two-field resolution using the HIBAG algorithm [46] and the population-specific Finnish and the multi-population 1000 Genomes references. The models were trained in seven different data compositions to evaluate the effect of model parameters and differences between the reference and target populations on model performance (Table 1). Imputation models for *HLA-G* and *HLA-G* 3'UTR and 5'UTR haplotypes were trained in four-field resolution to capture all the functionally relevant variability [58]. Since allele typing in four-field resolution was only available for the Finnish reference, these models could only be trained using the Finnish data. Numbers of individuals with available two-field or four-field resolution typing result in the reference data sets are summarized in Table 2, and the outline of the work is presented in Fig 1. Alleles present in the references are listed in Tables A and B in S1 Text.

The genotyping data for the Finnish reference was produced using FinnGen ThermoFisher Axiom custom array v2 (FIN I) or by full MHC genome sequencing (FIN II) and included 46,057 and 41,837 SNPs, respectively, within the MHC region. The WES-derived 1000 Genomes SNP data included 112,672 SNPs in the MHC region. For reference data compositions II-VI the Finnish and the 1000 Genomes SNP data was combined, and the SNPs shared between the datasets were used (Table 1). HIBAG recommends the use of flanking region of 500 kb for imputations. As the SNP arrays used for genotyping the references had a high SNP density, smaller flanking regions, 1 kb– 15 kb and 50 kb, were evaluated (Fig A in S1 Text). Since increasing the flanking region size to 50 kb did not improve the out-of-bag (OOB) or test accuracy, flanking region of 10 kb was chosen for all genes, and the models with 10 kb window size were used in the follow-up analyses.

**Table 1. Imputation models evaluated in the present study.**

| Model | Reference | SNP set | SNPs within MHC (FIN I/FIN II) | Allele calling method |
|---|---|---|---|---|
| I | Finnish | FIN | 46,057/41,837 | Clinical-grade |
| II | Finnish | FIN ∩ 1000G | 38,463/36,102 | Clinical-grade |
| III | 1000 Genomes European (EUR) | FIN ∩ 1000G | 38,463/36,102 | WES |
| IV | 1000 Genomes EUR and Finnish | FIN ∩ 1000G | 38,463/36,102 | Clinical-grade and WES |
| V | 1000 Genomes | FIN ∩ 1000G | 38,463/36,102 | WES |
| VI | 1000 Genomes and Finnish | FIN ∩ 1000G | 38,463/36,102 | Clinical-grade and WES |
| VII | 1000 Genomes | 1000G | 112,672 | WES |

FIN I; SNP data produced using the FinnGen ThermoFisher Axiom custom array v2

FIN II; SNP data produced by full MHC sequencing.

WES, short-read whole exome sequencing

## Overall accuracy of the imputation models

The overall accuracies of the imputation models were evaluated by comparing the imputed results with the sequence-based typing results and calculating the number of correctly predicted alleles over total number of alleles, as provided by the HIBAG accuracy statistics output. The overall cross-validation accuracies in all model compositions and test populations are summarized in Fig 2 and cross-validation confusion matrices for all models and test populations are shown in S2 Text. Model properties and OOB accuracies of the models trained with training and all reference data are presented in Fig B in S1 Text.

**Table 2. Numbers of individual samples with allele typing results per gene and reference data set.**

| | | Finnish | Reference | | | | | |
|---|---|---|---|---|---|---|---|---|
| | | | 1000 Genomes | | | | | |
| | | | EUR | AFR | EAS | SAS | AMR | 1000G all |
| MICA | Train | 512 | 219 | 314 | 145 | 208 | 165 | 1051 |
| | Test | 249 | 108 | 129 | 78 | 109 | 80 | 504 |
| | Total | 761 | 327 | 443 | 223 | 317 | 245 | 1555 |
| MICB | Train | 512 | 225 | 320 | 161 | 206 | 165 | 1077 |
| | Test | 249 | 110 | 148 | 78 | 103 | 90 | 529 |
| | Total | 761 | 335 | 468 | 239 | 309 | 255 | 1606 |
| HLA-E | Train | 295 | 315 | 340 | 311 | 208 | 186 | 1360 |
| | Test | 146 | 163 | 144 | 156 | 112 | 102 | 677 |
| | Total | 441 | 478 | 484 | 467 | 320 | 288 | 2037 |
| HLA-F | Train | 142 | 205 | 246 | 114 | 187 | 112 | 864 |
| | Test | 69 | 91 | 129 | 57 | 104 | 48 | 429 |
| | Total | 211 | 296 | 375 | 171 | 291 | 160 | 1293 |
| HLA-G | Train | 296 | | | | | | |
| | Test | 139 | | | | | | |
| | Total | 435 | | | | | | |
| HLA-G UTR | Train | 293 | | | | | | |
| | Test | 142 | | | | | | |
| | Total | 435 | | | | | | |

EUR, European; AFR, African; EAS, East Asians; SAS, South Asians; AMR, Mixed American

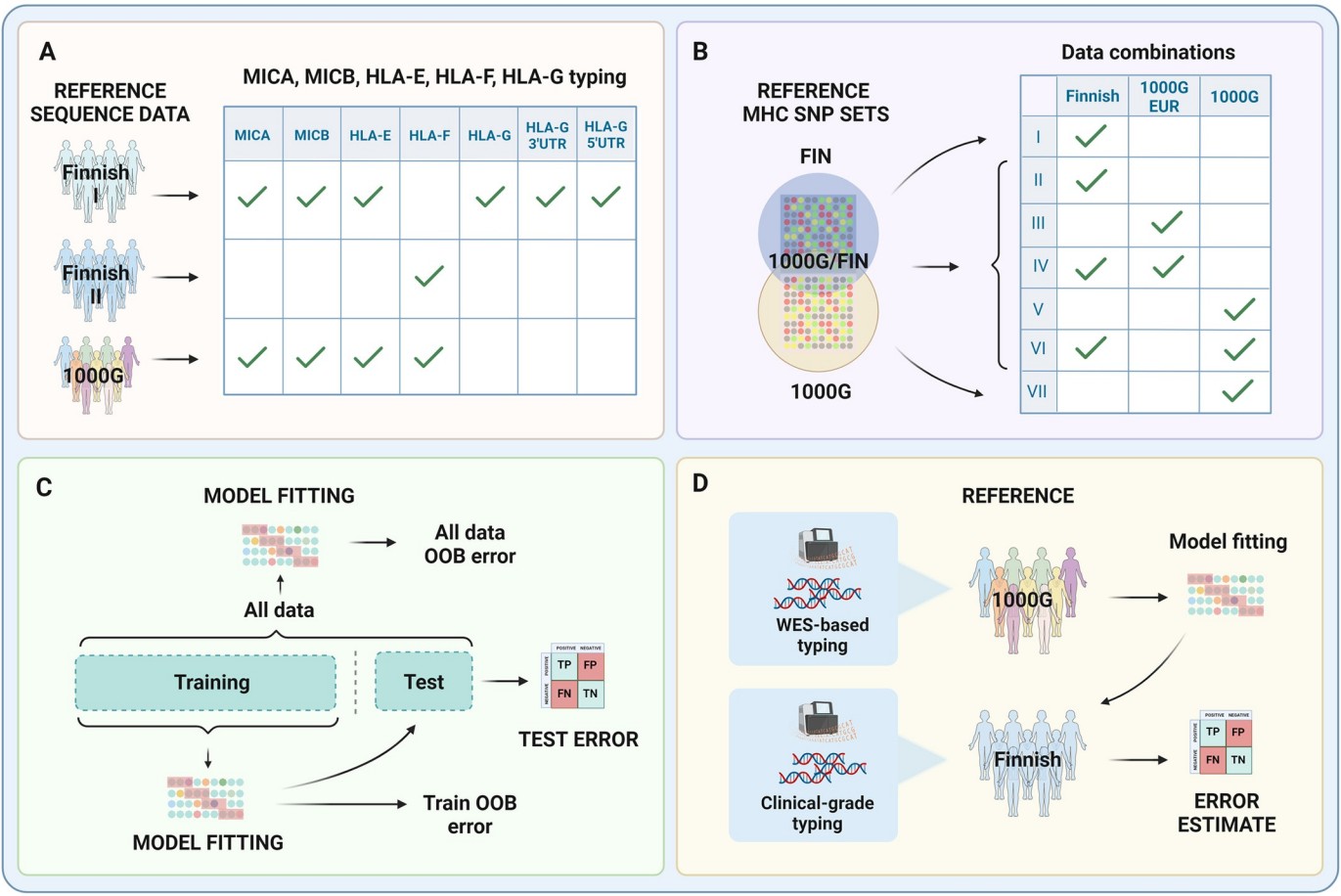

**Fig 1. Overall study design and workflow.** A) The reference data sets and genes sequenced and typed from each set; B) The reference data sets and their combinations for each imputation model (I–VII). The Venn diagram indicates the SNP contents used by the models: model I was based only on the SNPs in the Finnish reference set while model VII was based only on the 1000 Genomes SNPs. Models II—VI were based on the intersection of SNPs present on both reference sets. C) Evaluation of the model performance in an independent test set and out-of-bag (OOB) sets within the training and full data sets. D) Cross-validation of short-read whole exome sequencing (WES)-based allele calling in the 1000 Genomes reference against the clinical-grade typed Finnish reference. Created with BioRender.com.

When comparing the models trained with the different data compositions and the shared intersect SNPs (models II-VI), the overall accuracy increased as the size and diversity of the reference data increased (Fig 2). The overall accuracies averaged over all test populations and genes for models II-VI were (mean [min, max] (%) 95.9 [82.7, 100], 97.2 [86.4, 100], 97.8 [87.2, 100], 99.2 [97.4, 100] and 99.3 [97.4, 100], respectively, showing that the best overall accuracies were obtained for the 1000 Genomes reference (model V) or the combination of the 1000 Genomes and Finnish references (model VI).

The accuracy was evaluated also separately for the Finnish and the 1000 Genomes European (EUR), African (AFR), East Asian (EAS), South Asian (SAS) and Admixed American (AMR) superpopulations. The Accuracy in the Finnish population was consistently high, 99.8–100%, in all reference compositions for *MICA*, *HLA-E* and *HLA-F* (Fig 2). Accuracy was high also for *MICB* when using the Finnish reference-based model (II) (99.8%) but lower when using the 1000G EUR (III) or whole 1000 Genomes reference-based model (V) (97.8%) due to the *MICB*039* allele present in the Finnish reference but missing from the 1000 Genomes reference (S2 Text).

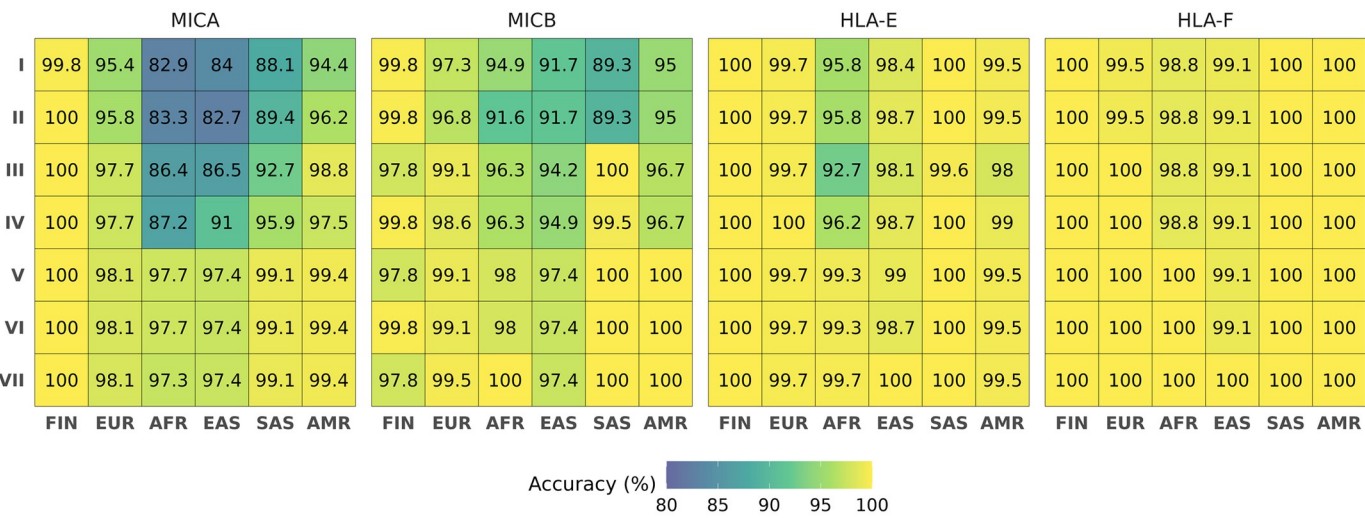

**Fig 2. Overall imputation accuracies of different models in each population and gene.** The horizontal axis shows the Finnish and the 1000 Genomes superpopulations (FIN, Finnish; EUR, European; AFR, African; EAS, East Asian; SAS, South Asian; AMR, Mixed American). The models trained on different reference compositions (I-VII) are shown on the vertical axis. Note that *HLA-G* is excluded because it is limited to the Finnish population only.

In the 1000 Genomes superpopulations, accuracies were lower for *MICA* and *MICB* when applying the Finnish reference (model II) but increased when applying the 1000G European (III) or combined 1000G European and Finnish (IV) reference. The accuracies were highest with the 1000 Genomes (V) or combined 1000G and Finnish (VI) references (Fig 2). In *HLA-E* and *HLA-F*, the differences between the data compositions were not as apparent and the accuracies were similar between the reference compositions, except for the clearly lower accuracies for AFR and AMR populations when using the 1000G EUR (III) reference only.

The overall accuracies shown in Fig 2 include all alleles present in the test set and imputation errors thus include both wrongly predicted alleles and untrained alleles, *i.e.*, alleles that are present in the test set but not present in the model. To examine the extent to which accuracy was affected by the untrained alleles in the test datasets, we calculated the overall accuracies by limiting only to the alleles present in the models (Fig C in S1 Text). The differences in percentage point were on average 0.009 with the highest difference for *MICA* (0.167) when applying the Finnish (II) reference-based model to the African population (Fig D in S1 Text). However, there were no differences between the two approaches with the best performing model (VI), so the accuracies achieved using this model were not affected by untrained alleles.

Based on the results of the model comparisons, we found model VI built on combined 1000 Genomes and Finnish reference (model VI) to be the best performing model.

## Performance of the best model

The best performing model based on model comparison was model VI trained on the combined 1000 Genomes and the Finnish reference. Accuracies averaged over all test populations for the genes *MICA*, *MICB*, *HLA-E* and *HLA-F* reached 98.6 [97.4, 100], 99.1 [97.4, 100], 99.5 [98.7, 100] and 99.9 [97.4, 100], respectively. The overall accuracies averaged over all genes reached 99.95 in the Finnish population and the accuracies in the different 1000 Genomes superpopulations were 99.2 [98.1, 100], 98.8 [97.7, 100], 98.2 [97.4, 99.1], 99.8 [99.1, 100] and 99.7 [99.4, 100] in EUR, AFR, EAS, SAS and AMR superpopulations, respectively, showing that the most challenging to impute were the African and East Asian populations (Fig 2 and S2

Text). The lowest overall accuracy was observed for *MICA*, ranging from 97.4% in EAS to 99.4% in AMR populations, whereas the highest accuracy was observed for *HLA-F* (99.1% in EAS and 100% in the rest of the populations), except for the Finnish population in which accuracy was consistently high (>99.8%) for all genes.

Rare alleles, such as *MICA*068:01*, *MICB*006*, *MICB*024:01* and *HLA-E*01:11*, were often erroneously imputed and had low sensitivity (Figs 3A and 4, and E in S1 Text, and Table C in S1 Text). The median level and interquartile range (IQR) of posterior probabilities (PP) producing zero erroneously imputed alleles for *MICA*, *MICB*, *HLA-E* and *HLA-F* were 0.992 (0.019), 0.997 (0.008), 0.997 (0.007) and 0.999 (0.001), respectively. The diagnostic accuracies of PP as a predictor of imputation errors measured by the receiver operating characteristic (ROC) area under the curve (AUC) for *MICA*, *MICB*, *HLA-E* and *HLA-F* were 0.752, 0.906, 0.942 and 0.986, respectively.

### The effect of SNP content

We also evaluated the effect of the number and content of SNPs on imputation accuracy by comparing the cross-validation results when using all SNP markers present in the reference data with the results when limiting the SNP selection to the markers common to both datasets (models I and II for the Finnish reference and models V and VII for the 1000 Genomes reference). The effect was more apparent in the 1000G reference that has a higher SNP density than the Finnish reference and thus was affected more by the intersection (56% vs. 9% reduction in model SNP number), but the differences in accuracy were modest and inconsistent between the genes and populations (Fig 2).

### HLA-G

The models for *HLA-G* alleles and *HLA-G* 3'UTR and *HLA-G* 5'UTR haplotypes were trained at four-field resolution to capture functionally relevant variation, since most of the variation is located in the untranslated regions of the gene affecting gene expression levels and post-transcriptional mRNA stability [58]. Thus, the models could only be trained using the Finnish reference (model I), for which four-field resolution typing results were available. Imputation accuracies of the models in the Finnish population reached 98.6%, 100% and 99.3% for *HLA-G*, 3'UTR and 5'UTR, respectively. As with the models for *MICA*, *MICB*, *HLA-E* and *HLA-F*, the correlation between the imputed and true allele dosages was lower for the low frequency alleles and UTR haplotypes, such as 5'UTR-other, *G*01:21N* and *G*01:01:14* (Fig E in S1 Text). The median level and IQR of posterior probabilities producing zero erroneously imputed alleles were 0.995 (0.010), 0.999 (5.293e-07) and 0.999 (6.920e-06) for *HLA-G*, 3'UTR and 5'UTR, respectively. The diagnostic accuracies of PP as a predictor of imputation errors measured by the AUC were 0.965, 1 (all imputation results were correct), and 0.761 for *HLA-G*, 3'UTR and 5'UTR, respectively. Overall test accuracies and model properties for *HLA-G*, *HLA-G* 3'UTR and *HLA-G* 5'UTR are summarized in Table 3 and confusion matrices are shown in Fig 3. Detailed allelic results are presented in Table D in S1 Text.

### Estimation of the accuracy of allele-calling in the 1000 Genomes reference

Since the allele assignment in the 1000 Genomes reference was done by short-read sequencing-based typing of the downloaded phase 3 full exome sequencing data and was susceptible to allele calling errors, we evaluated the correctness and quality of the 1000G reference by applying the model VII (*i.e.*, trained using 1000 Genomes reference) to the Finnish reference with clinical-grade typing quality. The overall accuracies as estimated by cross-validation were 99.6%, 99.8%, 100.0% and 99.8% for *MICA*, *MICB*, *HLA-E* and *HLA-F*, respectively. However,

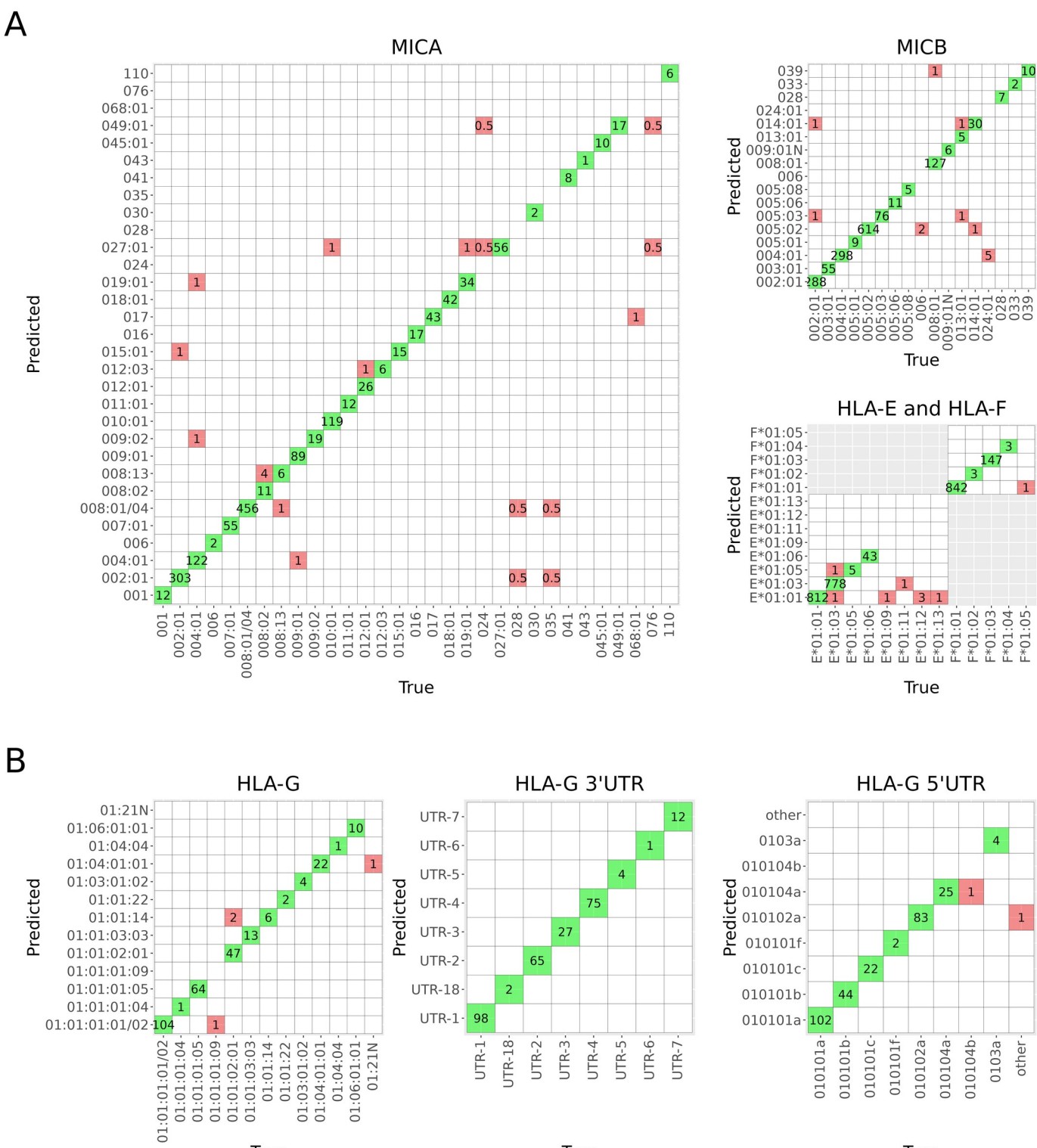

**Fig 3. Confusion matrices summarizing the allelic accuracies of the best gene-specific models.** A) The combined 1000G and Finnish reference (model VI) for *MICA*, *MICB*, *HLA-E* and *HLA-F*. B) The Finnish reference (model I) for *HLA-G*, *HLA-G* 3'UTR and *HLA-G* 5'UTR.

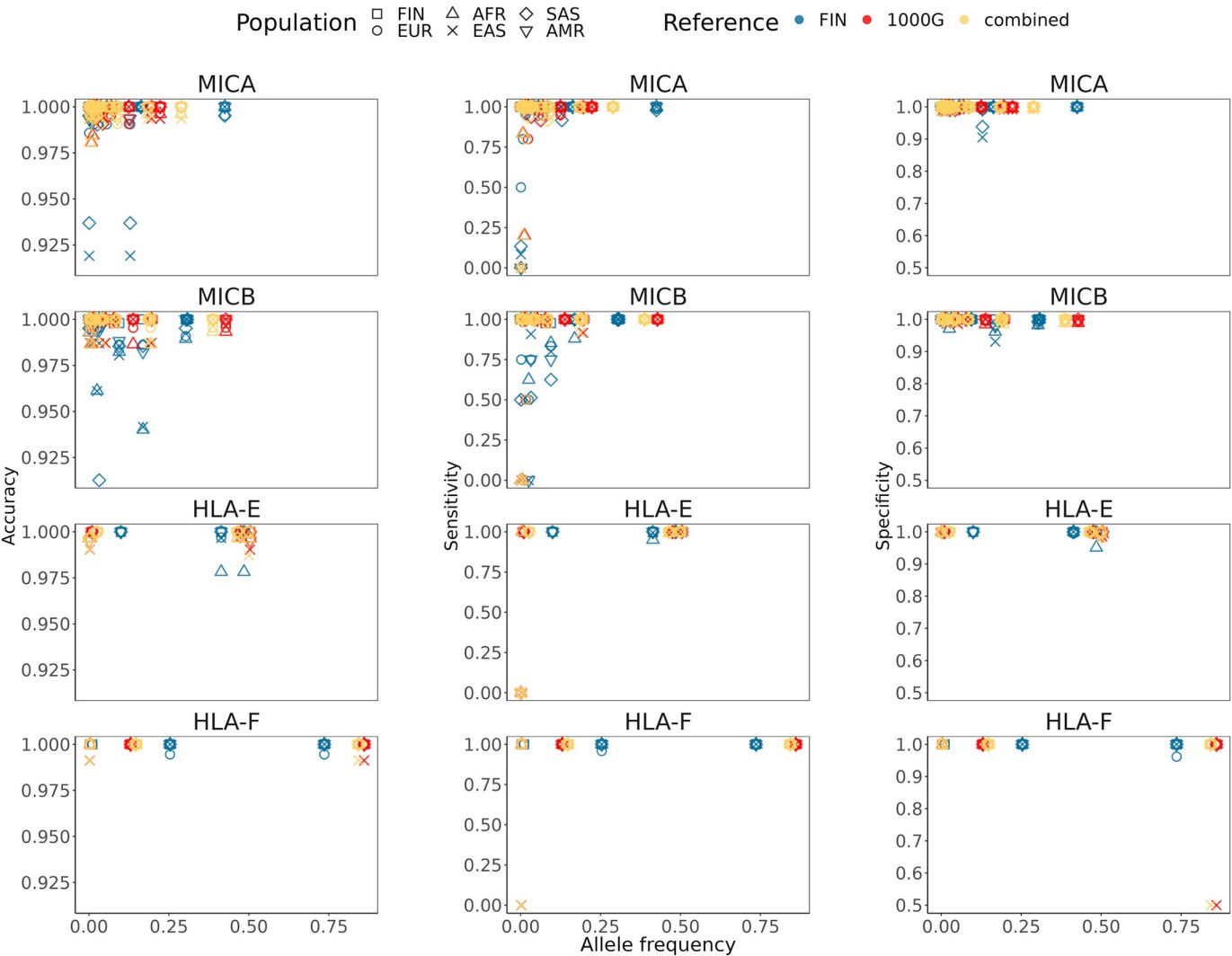

**Fig 4. The relationship between allele frequency and accuracy, sensitivity, and specificity.** The relationship is shown for the Finnish (FIN) and 1000 Genomes superpopulations (EUR, European; AFR, African; EAS, East Asian; SAS, South Asian; AMR, Mixed American) when using the Finnish (model II), 1000 Genomes (model V) and combined 1000G and Finnish (model VI) references in the training of the models. Notice the different y-axis scales in the panels.

this evaluation could only be done for the alleles that were shared by both references, thus the accuracy of the alleles, mostly rare, that were present in the 1000 Genomes reference but missing from the Finnish reference could not be assessed. Plot of the correlation between imputed and true allele dosages vs. allele frequency is presented in Fig E in S1 Text. The confusion matrices indicating allele-specific results and the alleles shared or missing between the references are presented in Fig 5.

**Table 3. Overall imputation accuracy and model parameters of *HLA-G* gene, *HLA-G* 3'UTR and *HLA-G* 5'UTR models.**

| Model | Training n | SNPs | Alleles/ haplotypes | OOB train | OOB all data | Test accuracy FIN |
|---|---|---|---|---|---|---|
| HLA-G | 296 | 450 | 20 | 97.2 | 98.1 | 98.6 |
| HLA-G 3'UTR | 293 | 450 | 9 | 99.6 | 99.7 | 100.0 |
| HLA-G 5'UTR | 293 | 453 | 10 | 99.3 | 99.4 | 99.3 |

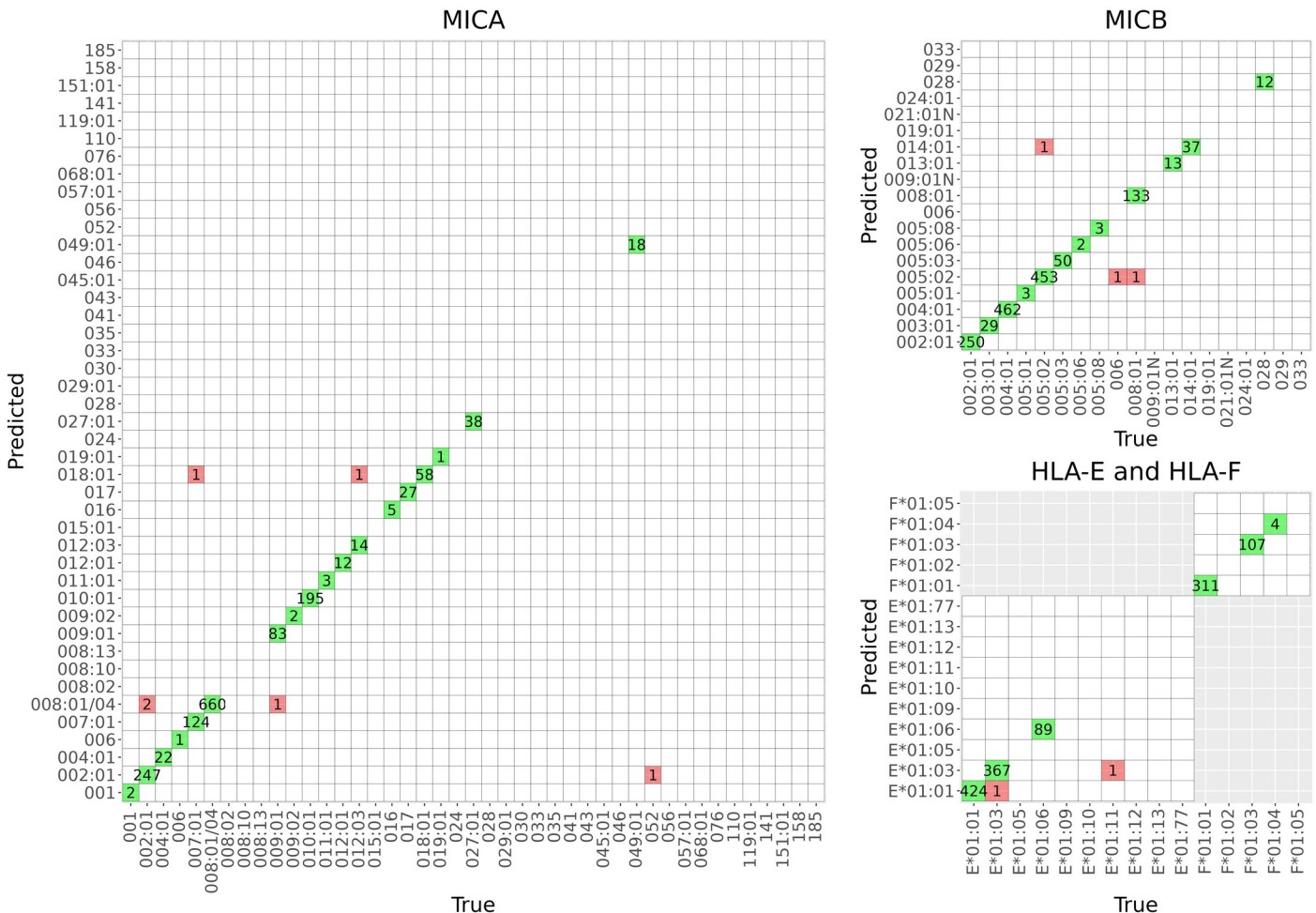

**Fig 5. Evaluation of the quality of the 1000 Genomes whole exome short-read sequencing-based allele calling.** Model VII was trained using the 1000 Genomes reference and applied to the Finnish reference with clinical-grade typing quality. Confusion matrices for *MICA*, *MICB*, *HLA-E* and *HLA-F* show the alleles that are common to both references and the amount of correctly and wrongly predicted alleles. Empty lines represent the alleles present in the model but absent from the Finnish reference that could not be validated. Overall imputation accuracies were 99.6%, 99.8%, 100.0% and 99.8% for *MICA*, *MICB*, *HLA-E* and *HLA-F*, respectively.

## Overall accuracy of imputation models fitted for GSA and PMRA SNP content

Since high density SNP data are not always available, we also fitted the imputation models to accommodate less SNP-dense genotyping array data. For this purpose, we selected two commonly used genotyping arrays, Infinium Global Screening Array (GSA, Illumina, Inc.) and Axiom Precision Medicine Research Array (PMRA, Thermo Fisher Scientific Inc.) and trained the models using the markers shared between the combined FIN I and 1000 Genomes/FIN II and 1000 Genomes reference and the GSA or PMRA array. The resulting intersection yielded 6955/7040 markers for the GSA model and 6871/7296 markers for the PMRA model. The models for *HLA-G*, *HLA-G* 3'UTR and *HLA-G* 5'UTR were trained using only the Finnish reference (FIN I) and the 7002 or 6930 common markers between the Finnish reference and the GSA or PMRA array, respectively. Since the intersected data were significantly less SNP-dense than the original reference data, a default flanking region of 500 kb recommended by HIBAG was used in the training of the models fitted for GSA and PMRA SNP content.

The overall accuracies averaged over all test populations for the models fitted for GSA SNP content were 97.2 [93.0, 100] for *MICA*, 99.0 [97.4, 100] for *MICB*, 99.3 [98.3, 100] for *HLA-E*, 99.6 [98.9, 100] for *HLA-F*, 99.3 for *HLA-G*, 100 for *HLA-G* 3'UTR and 99.6 for *HLA-G* 5'UTR (Fig F in S1 Text). Compared to the original models, the overall accuracies showed an average reduction of -0.4 percentage units. The most significant decrease was observed for *MICA* in the AFR population (-4.7).

The overall accuracies averaged over all test populations for the models fitted for PMRA SNP content were 97.4 [94.2, 100] for *MICA*, 98.8 [97.4, 99.8] for *MICB*, 99.1 [97.9, 99.7] for *HLA-E*, 99.6 [99.1, 100] for *HLA-F*, 98.6 for *HLA-G*, 100 for *HLA-G* 3'UTR and 98.9 for *HLA-G* 5'UTR (Fig F in S1 Text). Again, the average reduction in overall accuracies compared to the original models was -0.4 percentage units, with the most pronounced decrease observed for *MICA* in the AFR population (-3.5).

## Discussion

In the present study we have constructed and validated imputation models for the alleles of *MICA* and *MICB*, as well as those of the non-classical HLA genes *HLA-E*, *-F*, and *-G*, using both population-specific Finnish and a multi-population 1000 Genomes data as reference. To the best of our knowledge, this is the first study demonstrating an accurate stand-alone tool for imputing alleles in these immunologically relevant genes.

We evaluated the effects of population differences and model parameters to imputation accuracy by training the models using different reference data compositions. The highest over-all accuracies were achieved using the models V-VI containing the largest number of training samples and alleles. The significance of matching the reference and target populations to imputation accuracy has been demonstrated in previous studies [10,50–52] and was observed also in our results, as all the 1000 Genomes superpopulations showed significantly increased accuracies when applying the 1000 Genomes reference-based models compared with the Finnish reference-based model, especially for the more polymorphic *MICA* and *MICB*. For example, the *MICA* imputation of African (AFR) and East Asian (EAS) data reached an accuracy of ~97–98% when using the models trained with 1000 Genomes reference with or without the Finnish reference, but was significantly lower, ~83%, when using the model trained with the Finnish reference only. Even though we did not observe similar differences in the accuracies when imputing the Finnish population using the models trained with the Finnish or the 1000 Genomes European references, the effect was apparent in the *MICB\*039* allele that was only present in the Finnish reference and correctly imputed with the Finnish reference but missed when the 1000 Genomes reference was used. In concordance with previous studies [47,48,55], low frequency alleles were more difficult to impute correctly resulting in low prediction sensi-tivity as the models were not able to detect them in independent data partitions. Thus, the missed rare alleles were typically not reflected much in the overall accuracy estimates used to compare the models, which may bias the model evaluation towards the good performance with the more frequent alleles.

We also investigated the effect of SNP count on model performance. When the SNP content was reduced by 56% due to the intersection of reference data sets, we did not observe signifi-cant or systematic differences in accuracy, suggesting that this method is robust to variations in SNP content. However, it is important to note that the SNP density of our data was high. Consequently, the effect of SNP reduction could be much more apparent when using a less SNP-dense data. The clearest effects in accuracy were detected in alleles that differed by only one SNP (such as *MICB\*004:01* versus *024:01* or *HLA-E\*01:01* versus *01:12*), indicating that these alleles may pose challenges for accurate imputation using less SNP-dense array data. To

address this, we developed additional models tailored for the commonly used genotyping arrays, Infinium Global Screening Array (GSA, Illumina, Inc.) and Axiom Precision Medicine Research Array (PMRA, Thermo Fisher Scientific Inc.). These models reached >97% accuracy in all genes and test populations, except for the moderately lower accuracies in *MICA* for the African and East Asian populations, indicating that less SNP-dense genotyping data can still yield useful imputation results.

Imputation performance is highly dependent on the quality of the reference data. To build the multi-population reference, we used publicly available 1000 Genomes project to call alleles from short-read WES data. Many samples in the 1000 Genomes data had missing data or low read depth. Samples with no sequencing reads in *MICA*, *MICB*, *HLA-E*, *-F*, or *-G* area or too low read depth had to be discarded. Furthermore, to ensure reliability of the typing results, samples with a putative novel allele or an ambiguous typing result due to phasing ambiguities were also excluded from the reference. This may cause bias to the reference and its allele frequencies as the real variation is larger and is not detected by the models, which may skew the reference towards the more robustly decipherable genotypes. The method is also not able to detect the rare *MICA* whole-gene deletions and duplications with varying prevalence in different populations [59]. The Finnish reference is also limited by the relatively small number of individuals included in the panel, especially for the more polymorphic *MICA*, *MICB* and *HLA-G*. A larger independent test set would therefore have given a better estimate of the ability of the models to detect true variation in different populations. Hence, increasing the size and diversity of the multi-population reference would help build imputation models that better capture the true allele and haplotype heterogeneity in different populations.

Since the 1000 Genomes reference is not typed with clinical-grade quality, errors in the sequence analysis are still possible. We assessed the quality of the 1000 Genomes reference by applying a model trained on it to the Finnish reference data with clinical-grade typing results and achieved a high concordance, suggesting that allele calling errors in the 1000 Genomes reference are very rare. However, this assessment only included the alleles present in both references, hence, for some rare alleles no accuracy could be confirmed.

The imputation accuracies achieved in the present study are high enough for screening of genetic associations in large, genotyped cohorts, especially as the posterior probabilities are informative about the quality of imputation. With accurate imputation of the non-classical HLA and MIC genes from large cohorts it is possible to detect novel associations and to study traits related to transplantation, cancer, autoimmunity, mother-fetus interface and NK cell activities, in which these molecules have functional roles [60–65]. The method is also of use for fine mapping traits whose MHC associations at the SNP level have indicated signals outside the classical HLA genes and can help clarify whether these SNPs are markers for allelic variants of the non-classical or MIC genes. Previous association studies focusing on MHC region SNPs have suggested that the success of unrelated HSCT might be influenced by non-HLA genetic variation within the MHC, for example Petersdorf et al [38] identified two SNPs as markers for disease-free survival and acute GVHD, one of which is a putative expression quantitative locus for *MICA* and *MICB* genes. Similar studies have also identified SNPs in the MHC class I region or close to *MICA* and *MICB* that associate with multiple sclerosis and psoriasis [41,42]. Fine-mapping the MHC region near the SNPs of interest can help identify the true causal genetic variants conferring the risks and examine the potential mechanisms involved in disease pathogenesis.

The imputation models of this study are available at Github (https://github.com/FRCBS/HLA_EFG_MICAB_imputation) and can be readily installed and ran in local computers without the need of sending individual genetic data sets to remote portals, a procedure often forbidden by legislation or patient consents.

## Materials and methods

### Ethics statement

The study was carried out in accordance with the permits granted by the Ethical Review Board of the Hospital District of Helsinki and Uusimaa (decision HUS_1252_2020) and Turku (decision TYKS_ETMK_28_2012).

The use of genetic data of the Blood Service Biobank of the Finnish Red Cross Blood Service is in accordance with the biobank consent of the sample donors and meets the requirements of the Finnish Biobank Act 688/2012. The study protocol was accepted by the Blood Service Biobank (decision 002–2018). Biobank samples for genotyping were collected only from blood donors who gave a written biobank consent.

### Finnish reference

A dataset of Blood Service Biobank blood donors (Finnish I) with genomic SNP array and targeted sequencing data was used as a reference for *MICA* (n = 761), *MICB* (n = 761), *HLA-E* (n = 441) and *HLA-G* (n = 435). SNP array data including 46,057 SNPs within the MHC region was produced using the FinnGen ThermoFisher Axiom custom array v2 [66]. The genotyping and genotype imputation procedure is described in detail in Tabassum et al [67].

Targeted sequencing of *MICA*, *MICB*, *HLA-E* and *HLA-G* genes was acquired from Histogenetics (Histogenetics, Ossining, NY 10562, USA), and produced using PacBio long read sequencing platform. Allele assignment was done at two-field (*i.e.*, defining protein sequence level variation) resolution for *MICA*, *MICB* and *HLA-E* and at four-field resolution (*i.e.*, defining nucleotide sequence level variation) for *HLA-G* using NGSengine software v2.21.0 (GenDx, Utrecht, The Netherlands) and IMGT/HLA database nomenclature release v3.42 and by manual inspection of read alignments when necessary. In total, 761 samples with *MICA* and *MICB*, 441 with *HLA-E* and 435 with *HLA-G* typing were available. The number of different *MICA*, *MICB*, *HLA-E* and *HLA-G* alleles present in the data set were 22, 19, 4 and 23, respectively. Two novel alleles of *MICA* and two of *MICB* were detected [68], as well as four novel alleles of *HLA-G* (01:L14P, 01:01:01:g.173G/A, 01:01:01:g.188C/T and 01:01:01:g.636C/T). *HLA-G* 3' UTR and 5' UTR haplotypes were also inferred from the SNP genotype data using as a reference the 3' UTR and 5' UTR haplotypes as defined by Castelli et al [69]. Haplotypes that did not correspond to the defined haplotypes were classified as 'other'.

Since targeted *HLA-F* sequencing was not available commercially, 211 apparently healthy Finnish individuals who were genotyped by full MHC genome sequencing at the McGill Genome Centre [70] (McGill University, Montreal, Canada) were used as a reference for HLA-F (Finnish II). The SNP data included 41,837 SNP markers in the MHC region [71]. *HLA-F* allele assignment at two-field resolution was performed using Omixon Explore software v2.0.0 (Omixon, Budapest, Hungary) with IMGT/HLA database nomenclature release v3.42 and by manual inspection of read alignments when necessary. Three different *HLA-F* alleles were present in the reference dataset.

### 1000 Genome Project reference

1000 Genomes Project phase 3 data was used as reference for the multi-population imputation models for *MICA*, *MICB*, *HLA-E* and *HLA-F*. SNP-array and full exome sequencing data of 1000 Genomes Project phase 3 [72] was downloaded from https://ftp.1000genomes.ebi.ac.uk/vol1/ftp/data_collections/1000_genomes_project/data/ as cram files and processed with samtools v1.14 *view -bT* command [73] to extract specific gene regions using GRCh38_full_analysis_set_plus_decoy_hla.fa sequence file as a reference. Fastq reads in *MICA*, *MICB*, *HLA-E* and

*HLA-F* loci were extracted from bam files using Bazam v1.0.1 [74] and allele assignment was done at two-field resolution using NGSengine software v.2.21.0 (GenDx, Utrecht, The Netherlands) and IMGT/HLA database nomenclature release 3.42 and by manual inspection of read alignments when necessary. The number of *MICA*, *MICB*, *HLA-E* and *HLA-F* alleles in the final 1000 Genomes reference dataset were 43, 19, 10 and 5, respectively.

1000 Genomes SNP data included 112,672 SNPs in the MHC region. Liftover of the SNP positions from Genome Reference Consortium Human Build 37 (GRCh37) to build 38 (GRCh38) was done using NCBI Remap (https://www.ncbi.nlm.nih.gov/genome/tools/remap). Numbers of individuals with available two-field or four-field resolution typing result in the reference data sets are summarized in Table 2. Alleles present in the references are listed in Tables A and B in S1 Text.

## Model training and validation

We trained the Finnish and multi-population imputation reference models using the R-package HIBAG [46], version 1.38.1, that allows building of models with custom references. The models were trained with seven different data compositions (Table 1 and Fig 1) to evaluate the effect of model parameters (Fig B in S1 Text) and the differences in ethnicity between the reference and target populations on imputation accuracy.

Models I and VII were trained using the Finnish reference and the 1000 Genomes reference, respectively. For models IV and VI, the Finnish and the 1000 Genomes references were combined with HIBAG 'hlaGenoCombine' function with default parameters to create a mixed reference and the SNPs in the MHC region that were common to both genotyping data (38,463 between Finnish I and 1000 Genomes data and 36,102 between Finnish II and 1000 Genomes data) were used in the building of the models. In models II, III and V, the same intersect markers as in IV and VI were used for fair comparisons of the models. The SNPs in the gene area, as defined by the HIBAG 'hlaFlankingSNP' function with 'assembly = hg38' parameter, and flanking regions 1 kb– 15 kb and 50 kb, were evaluated (Fig A in S1 Text) to determine the best SNP window size for the training of the models. The reference data was randomly partitioned into two thirds training set and one third test set for each gene (Table 2), with the exception that alleles present in the reference in less than 3 copies were distributed evenly between the training and the test sets. HIBAG imputation models with 100 classifiers were then fitted for each data composition and gene using the training portion of the data (training sets in Table 2) and HIBAG v1.38.1 imputation algorithm.

To evaluate the performance of the models, a cross-validation was performed using the one third proportions (test sets in Table 2) of the Finnish reference and the 1000 Genomes reference that were not used for the training of the models, using variant positions in genome build hg38. The accuracies were calculated for all loci by comparing the imputed alleles with the sequence-based typing results and counting the number of correctly predicted alleles divided by the number of all alleles. Finally, imputation models were trained using all reference data. SNPs included in the final models are listed in S1 Data. The overall study workflow is presented in Fig 1.

## Evaluation of the accuracy of allele calling in 1000 Genomes reference

The allele assignment in the 1000 Genomes reference was done by sequence-based typing of the downloaded phase 3 short-read full exome sequencing data. To evaluate the correctness of allele calling and quality of the reference, a cross-validation of the 1000 Genomes reference was performed by applying the imputation models trained with the 1000 Genomes reference (model VII) to the Finnish reference with clinical-grade typing results.

### Fitting of models for GSA and PMRA markers

The models were also trained for two commonly used, less SNP-dense, arrays Infinium Global Screening Array (GSA, Illumina, Inc.) and Axiom Precision Medicine Research Array (PMRA, Thermo Fisher Scientific Inc.) using the intersect markers of the combined Finnish (FIN I for *MICA*, *MICB* and *HLA-E* and FIN II for *HLA-F*) and 1000Genomes reference and the GSA or PMRA array. There were in total 8635 and 9563 markers within the MHC region in the GSA and PMRA arrays, respectively. Of these markers, 6955/7040 and 6871/7296 were common with the combined FIN I and 1000 Genomes/FIN II and 1000 Genomes reference and used in the training of the models. Models for *HLA-G*, *HLA-G* 3'UTR and *HLA-G* 5'UTR were trained using only the Finnish reference (FIN I) and the common markers between the Finnish reference and GSA or PMRA array (7002 and 6930 markers, respectively). All models were trained similarly to the Finnish/1000G models described in 'Model training and validation' except for a larger HIBAG default 5000 kb flanking region used to select the markers.

R code for training and validation of the imputation models are available in Github (https://github.com/FRCBS/HLA_EFG_MICAB_model_training).

## Supporting information

**S1 Text. Supplementary Tables and Figures.**
(PDF)

**S2 Text. Confusion matrices.**
(PDF)

**S1 Data. Model SNPs.**
(XLSX)

## Acknowledgments

We want to thank Blood Service Biobank of the Finnish Red Cross Blood Service for providing DNA samples and MHC region SNP data. We also want to thank Leila Taalikainen and Marko Haverinen for their help with sample processing.

## Author Contributions

**Conceptualization:** Silja Tammi, Satu Koskela, Jukka Partanen, Jarmo Ritari.

**Formal analysis:** Silja Tammi, Jarmo Ritari.

**Funding acquisition:** Jukka Partanen, Jarmo Ritari.

**Investigation:** Silja Tammi, Satu Koskela.

**Methodology:** Silja Tammi, Jarmo Ritari.

**Project administration:** Satu Koskela, Jukka Partanen.

**Resources:** Blood Service Biobank, Kati Hyvärinen.

**Supervision:** Jukka Partanen, Jarmo Ritari.

**Writing – original draft:** Silja Tammi, Jukka Partanen, Jarmo Ritari.

**Writing – review & editing:** Silja Tammi, Satu Koskela, Kati Hyvärinen, Jukka Partanen, Jarmo Ritari.

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
