## [Decision Letter · Decision Letter 0]

16 Apr 2024

Dear Ms Tammi,

Thank you very much for submitting your manuscript "Accurate multi-population imputation of MICA, MICB, HLA-E, HLA-F and HLA-G alleles from genome SNP data" for consideration at PLOS Computational Biology. As with all papers reviewed by the journal, your manuscript was reviewed by members of the editorial board and by several independent reviewers. The reviewers appreciated the attention to an important topic. Based on the reviews, we are likely to accept this manuscript for publication, providing that you modify the manuscript according to the review recommendations.

The manuscript has been seen by two expert reviewers, and both find the work very valuable for the community. The reviewers suggest some improvements that I would encourage the authors to follow. Both reviewers address to point to which extent this imputation approach can be used on data from SNP arrays rather than WGS/WES data, and it would be important to see at least a discussion and clarification of this. If the authors think that their approach would also work with such less dense SNP data, they are welcome to provide additional test as suggested by reviewer 2.

Sincerely,

Tobias Lenz

Guest Editor

PLOS Computational Biology

Mark Alber

Section Editor

PLOS Computational Biology

The manuscript has been seen by two expert reviewers, and both find the work very valuable for the community. The reviewers suggest some improvements that I would encourage the authors to follow. Both reviewers address to point to which extent this imputation approach can be used on data from SNP arrays rather than WGS/WES data, and it would be important to see at least a discussion and clarification of this. If the authors think that their approach would also work with such less dense SNP data, they are welcome to provide additional test as suggested by reviewer 2.

Reviewer's Responses to Questions

**Comments to the Authors:**

Reviewer #1: In this work, Silja Tammi et al. perform a thorough investigation and development of an imputation pipeline for MICA, MICB, HLA-E, HLA-1 F and HLA-G (including UTR polymorphism). The resources used are multiethnic and include the 1000 genomes and Finn Gen. I have only a few comments:

Having an idea of the allele frequency of the predicted alleles is important as imputation can be very accurate if polymorphisms are rare. This is indirectly presented in the figures with the counts, which I particularly appreciated, together with the availability of the data.

They could have further validated their algorithm using the UKB data

There are 2 limitations of this work, one clearly stated the other less so.

Then first limitation pertains to the fact a lot of incomplete coverage sequence had to be eliminated. This could create a bias, and possibly having an effect on rare alleles, but this is acknowledged.

The second limitation has to do with the scope of the study. The title and the abstract, unless read carefully, could be misinterpreted as if the authors were establishing a high accuracy imputation method for these gene alleles usable with whole genome association genotyping arrays. This is not the case, it is an imputation method to be used for exome or whole genome sequencing reads; this also explains the high accuracy as it uses dense SNP within the gene (+/- a few kb). In theory the whole coding information needed to define each allele is there except phasing in some cases where long reads could be needed.

It does not change the fact this is useful especially now that WGS and WES is mainstream and that questionable results such as in Commun Biol. 2023; 6: 1113 are starting to be published (the paper reports high frequencies of unknown alleles in the UKB). I would suggest that the title "Accurate multi-population imputation of HLA-E, HLA-F, HLA-G, MICA and MICB alleles from genome SNP data" be changed into "Accurate multi-population imputation of HLA-E, HLA-F, HLA-G, MICA and MICB alleles from genome SNP data derived from exome or whole genome sequencing".

Also, in the abstract, the sentence "To facilitate studies of the nonclassical and non-HLA genes in large patient and biobank cohorts, we trained imputation models for MICA, MICB, HLA-E, HLA-F and HLA-G alleles on genome SNP array data" could be changed into "To facilitate studies of the nonclassical and non-HLA genes in large patient and biobank cohorts with whole genome and exome sequencing, we trained and show highly accurate imputation models for MICA, MICB, HLA-E, HLA-F and HLA-G alleles base on genome SNP array data".

This limitation and the state of imputation of these gene alleles using whole genome SNP arrays like Affy PMRA or illunina infinium could have been discussed (or attempted since FinnGen has been genotyped with these). My recollection is that only Deep-HLA a deep learning based method developed by Naito and Okada in japan (Nature Communications volume 12, Article number: 1639 (2021)) exists but I am not sure how good it is for these loci. Finally, some of these genes (MICA, etc) have rare duplication and deletions, see Front. Immunol., 14 November 2023, duplication/deletions could be also mentioned as a limitation as I am not sure how these are handled.

Reviewer #2: Comments to the Authors:

In this manuscript (Tammi et al.), the authors implemented imputation models for MICA, MICB, HLA-E, HLA-F and HLA-G alleles in HIBAG framework. They used clinical-grade allelic information for Finnish dataset and also WES-based estimated alleles for 1KG dataset. They showed accuracy metrics for each of the models combining both sources. These models are publicly released through github. I think the models for imputing the non-classical genes within MHC would be valuable resource for the community. I have several major comments mostly for the metrics to assess the imputation performance, and for clarity of the description of imputation and validation methods.

1. The ‘accuracy’ metrics that the authors described are known to be upwardly biased for rare alleles by random chance. I would like to see the dosage correlation of the true alleles vs. imputed alleles as another way to assess the imputation performance, especially in the section of using 1KG as a reference model and Finnish data as the target of imputation and comparison with the clinical-grade ‘true’ alleles. It would also be useful to assess the dosage correlation as a function of minor allele frequency, which I believe is the standard of the field.

2. In the Result section (main text), I would recommend the authors to describe briefly about the Finnish dataset, including how the alleles were called, before moving onto the ‘Imputation model development’ section for readers to understand the overview of the samples used in the reference. Also, briefly describing the SNPs in the 1KG and Finnish datasets would be helpful.

3. Has the imputation performance been compared between flanking regions 50kb vs. 500kb (default of HIBAG)?

4. Considering realistic setting of the MHC imputation, it would be nice to assess the imputation performance using SNPs from commonly used genotyping array (e.g., Illumina MEGA/GSA). Using the SNPs from 1KG, the authors could create ‘mock’ genotyping data for these arrays (target) and assess the performance using Finnish-based model as a reference.

5. As another consideration for realistic setting of the MHC imputation, most of the investigators would be interested in fine-mapping the diseases alleles in the entire MHC region, not limiting to the MICA, MICB, HLA-E, HLA-F and HLA-G alleles. If the entire MHC region is imputed altogether (including classical HLA alleles), we can perform conditional analyses to identify the exhaustive combination of the MHC alleles independently associated with the disease. To this end, the reference models not only for the genes the authors focused on but also for the classical HLA alleles with the same reference samples should be very valuable. Would it be possible for the authors to release the models for the classical HLA alleles, or is it already publicly accessible?

**Have the authors made all data and (if applicable) computational code underlying the findings in their manuscript fully available?**

Reviewer #1: Yes

Reviewer #2: Yes

PLOS authors have the option to publish the peer review history of their article (what does this mean?). If published, this will include your full peer review and any attached files.

Reviewer #1: **Yes: **Emmanuel Mignot

Reviewer #2: No

Figure Files:

Data Requirements:

Reproducibility:

References:

---

## [Decision Letter · Decision Letter 1]

31 Aug 2024

Dear Ms Tammi,

We are pleased to inform you that your manuscript 'Accurate multi-population imputation of MICA, MICB, HLA-E, HLA-F and HLA-G alleles from genome SNP data' has been provisionally accepted for publication in PLOS Computational Biology.

Best regards,

Stacey D. Finley, Ph.D.

Section Editor

PLOS Computational Biology

Stacey Finley

Section Editor

PLOS Computational Biology

Reviewer's Responses to Questions

**Comments to the Authors:**

Reviewer #1: I have no comments on this revision.

Reviewer #2: I appreciate the author's thorough response and additional analyses based on my previous comments. I liked the revised manuscript including additional description of the markers and the SNP-array based evaluation. It was also nice to see high dosage correlation for most common alleles. I still think that combined reference panel for the paper's genes with conventional classical HLA genes in the same set of samples is most accurate for users aiming for joint fine-mapping, rather than separate panels and post-hoc integration after imputation as the authors mentioned. But I agree that this might be beyond the scope of this manuscript.

**Have the authors made all data and (if applicable) computational code underlying the findings in their manuscript fully available?**

Reviewer #1: Yes

Reviewer #2: Yes

PLOS authors have the option to publish the peer review history of their article (what does this mean?). If published, this will include your full peer review and any attached files.

Reviewer #1: No

Reviewer #2: No

---

## [Editor Report · Acceptance letter]

11 Sep 2024

PCOMPBIOL-D-23-01921R1 

Accurate multi-population imputation of MICA, MICB, HLA-E, HLA-F and HLA-G alleles from genome SNP data

Dear Dr Tammi,

I am pleased to inform you that your manuscript has been formally accepted for publication in PLOS Computational Biology. Your manuscript is now with our production department and you will be notified of the publication date in due course.

With kind regards,

Anita Estes
